# EEPO: Exploration-Enhanced Policy Optimization via Sample-Then-Forget

**Liang Chen**[1,5]    **Xueting Han**[2]    **Qizhou Wang**[3]    **Bo Han**[3]    **Jing Bai**[2]
**Hinrich Schütze**[4]    **Kam-Fai Wong**[1,5]

[1]The Chinese University of Hong Kong    [2]Microsoft Research Asia
[3]Hong Kong Baptist University    [4]LMU Munich
[5]MoE Key Laboratory of High Confidence Software Technologies

{lchen, kfwong}@se.cuhk.edu.hk    chrihan@microsoft.com

## Abstract

Balancing exploration and exploitation remains a central challenge in reinforcement learning with verifiable rewards (RLVR) for large language models (LLMs). Current RLVR methods often overemphasize exploitation, leading to entropy collapse, diminished exploratory capacity, and ultimately limited performance gains. Although techniques that increase policy stochasticity can promote exploration, they frequently fail to escape dominant behavioral modes. This creates a self-reinforcing loop—repeatedly sampling and rewarding dominant modes—that further erodes exploration. We introduce **E**xploration-**E**nhanced **P**olicy **O**ptimization (**EEPO**), a framework that promotes exploration via two-stage rollouts with adaptive unlearning. In the first stage, the model generates half of the trajectories; it then undergoes a lightweight unlearning step to temporarily suppress these sampled responses, forcing the second stage to explore different regions of the output space. This *sample-then-forget* mechanism disrupts the self-reinforcing loop and promotes wider exploration during rollouts. Across five reasoning benchmarks, EEPO outperforms GRPO, achieving average relative gains of 24.3% on Qwen2.5-3B, 33.0% on Llama3.2-3B-Instruct, and 10.4% on Qwen3-8B-Base.

## 1 Introduction

The emergence of OpenAI's o1 (OpenAI) and DeepSeek-R1 (DeepSeek-AI et al., 2025) marks a significant advance in LLM reasoning. A key driver of this progress is reinforcement learning with verifiable rewards (RLVR) (DeepSeek-AI et al., 2025), powered by the Group Relative Policy Optimization (GRPO) (Shao et al., 2024). Nevertheless, RLVR continues to face the classic exploration–exploitation dilemma (Sutton & Barto, 2018) due to the exploitative nature of its objectives. Specifically, policies tend to over-emphasize exploitation of high-reward trajectories, leading to entropy collapse and reduced final performance (Yu et al., 2025; Cui et al., 2025).

In this work, we examine entropy collapse on Qwen2.5-3B. We observe that as entropy declines sharply, in-distribution test accuracy continues to rise, whereas performance on out-of-distribution benchmarks (e.g., AMC 2023) deteriorates (Figure 2). This suggests reduced exploration drives overfitting to the training distribution rather than discovering generalizable reasoning patterns. We hypothesize that, as entropy falls, the policy forms increasingly confident beliefs about solutions, yielding a response distribution with multiple, imbalanced modes (Figure 3a): several plausible reasoning behaviors exist for a given question, but one mode receives more probability mass. If rollouts predominantly sample this dominant mode and receive positive feedback, the policy further amplifies it while suppressing alternatives (Figure 3b). This *self-reinforcing loop* accelerates entropy collapse. Crucially, it impedes the discovery of alternative—potentially superior—reasoning strategies, causing local optima and poor generalization.

---

[*]Work was done during Liang Chen's internship at MSRA.
[†]Corresponding to: Xueting Han and Kam-Fai Wong.

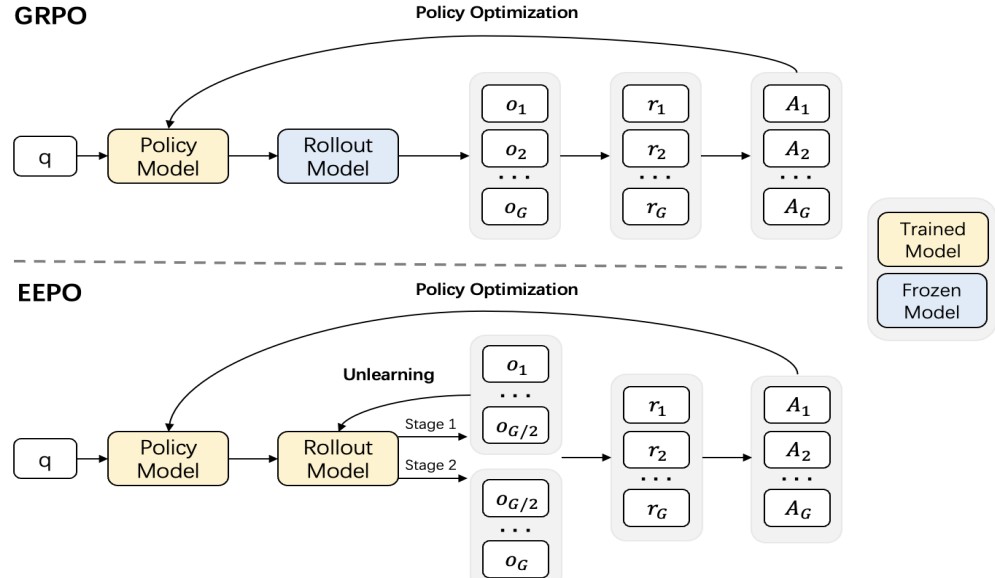

Figure 1: Comparison of GRPO and EEPO rollout processes. GRPO samples all trajectories from a fixed rollout model, while EEPO introduces an unlearning step on the rollout model between two sampling stages to promote exploration of diverse modes.

Recent efforts to improve exploration in RLVR largely fall into two categories: objective-level modifications and indiscriminate exploration. Approaches such as increasing the sampling temperature (Ziegler et al., 2019) or adding entropy regularization (Hou et al., 2025) flatten the output distribution uniformly (Figure 3c). While this increases stochasticity, it fails to shift probability mass away from dominant behaviors and often yields instability or degraded performance when applied aggressively (Figure 5). A widely adopted recent approach, DAPO (Yu et al., 2025), increases the upper clipping threshold to grant low-probability trajectories fewer restrictions during training. Yet these objective-level tweaks do not break the self-reinforcing loop: during rollouts, the policy remains confined to dominant modes and fails to explore beyond previously sampled high-probability regions.

To address this problem, we propose Exploration-Enhanced Policy Optimization (EEPO), a method that promotes exploration by preventing repeated sampling from dominant modes during rollout. Specifically, EEPO introduces a *sample-then-forget* mechanism that divides the GRPO rollout into two stages, as shown in Figure 1: the rollout model first generates half of the trajectories, then performs a temporary unlearning step to suppress the just-sampled responses. The remaining trajectories are sampled from this updated model. Unlike objective-level approaches, this mechanism operates directly within the rollout process, explicitly encouraging subsequent samples to deviate from dominant behaviors and uncover alternative trajectories—thereby steering exploration toward broader regions, as illustrated in Figure 4.

To adapt the unlearning intervention to RL exploration, we introduce three design choices that make it targeted, triggerable, and lightweight. First, to impose stronger penalties on dominant regions, we replace the standard negative log-likelihood with a complementary loss that penalizes high-probability tokens more than low-probability ones. Second, to trigger intervention at the onset of mode collapse, we introduce an entropy-conditioned gating mechanism that activates unlearning only when exploration deteriorates (i.e., low entropy). Finally, to keep the intervention lightweight and temporary, we apply a single-step gradient update to the GRPO rollout model—synchronized from the actor in each iteration and used solely for sampling—thereby decoupling unlearning from policy optimization and confining its effect to the rollout phase.

To validate our approach, we evaluate EEPO on five challenging mathematical reasoning benchmarks using three distinct LLMs. The benchmarks include Minerva Math (Lewkowycz et al., 2022), OlympiadBench (He et al., 2024), and three competition-level datasets: AMC 2023, AIME 2024, and AIME 2025. EEPO consistently outperforms the baselines, yielding average relative improvements over GRPO of 24.3% on Qwen2.5-3B, 33.0% on Llama3.2-3B-Instruct, and 10.4% on Qwen3-8B-Base. Furthermore, our analyses show that EEPO achieves superior performance through more effective exploration while maintaining comparable training time to standard GRPO. The code will be available at https://github.com/ChanLiang/EEPO.

## 2 PRELIMINARIES

We begin by reviewing RL with Verifiable Rewards (RLVR) (DeepSeek-AI et al., 2025) and its prevalent implementation, Group Relative Policy Optimization (GRPO) (Shao et al., 2024), which has been widely adopted for training large-scale reasoning models. We then analyze its limitations related to insufficient exploration and revisit existing solutions attempted to mitigate this issues.

### 2.1 RL FOR TRAINING LARGE-SCALE REASONING MODELS

**RLVR.** The success of RLVR relies on reliable reward signals (DeepSeek-AI et al., 2025), typically provided by a rule-based reward model that delivers precise feedback for tasks in mathematical, coding, and logical reasoning domains. Consider a mathematical dataset $\mathcal{D} := \{(q, a)\}$, where $q$ is a question and $a$ is its ground-truth final answer . The reward depends solely on the correctness of the final prediction $\hat{a}$ compared to $a$, without enforcing constraints on the reasoning process:

$$r(\hat{a}, a) = \mathbb{1}[\hat{a} \equiv a]. \tag{1}$$

The RLVR objective is often implemented using the large-scale policy optimization method GRPO. Compared to proximal policy optimization (PPO; Schulman et al., 2017), GRPO improves computational efficiency by eliminating the need for a separate value function.

**GRPO.** As illustrated in Figure 1, given a question $q$ and a set of responses, i.e., reasoning paths, $O = \{o_1, o_2, \ldots, o_G\}$ sampled from the old policy model $\pi_{\text{old}}$, GRPO directly computes advantages to optimize the policy model $\pi$ using the following objective:

$$\mathcal{J}_{\text{GRPO}}(\theta) = \frac{1}{\sum_{i=1}^{G} |o_i|} \sum_{i=1}^{G} \sum_{t=1}^{|o_i|} \min \left[ r_{i,t}(\theta) \hat{A}_i, \text{clip}\left(r_{i,t}(\theta), 1 - \epsilon, 1 + \epsilon\right) \hat{A}_i \right] - \beta \mathbb{D}_{\text{KL}}[\pi_\theta \parallel \pi_{\text{ref}}]. \tag{2}$$

Here, $\pi_{\text{ref}}$ denotes a reference model used to constrain policy updates via a KL divergence penalty. The score $\hat{A}_i$ represents the normalized advantage of response $o_i$, computed as $\hat{A}_i = \frac{r_i - \text{mean}(\{r_1, \ldots, r_G\})}{\text{std}(\{r_1, \ldots, r_G\})}$, where $\{r_1, \ldots, r_G\}$ denotes the rewards corresponding to the sampled responses in the group $O$.

The importance weight $r_{i,t}(\theta)$ denotes the probability ratio between current and old policies:

$$r_{i,t}(\theta) = \frac{\pi_\theta(o_{i,t} \mid q, o_{i,<t})}{\pi_{\theta_{\text{old}}}(o_{i,t} \mid q, o_{i,<t})} \tag{3}$$

This importance sampling ratio is crucial for obtaining *unbiased* gradient estimates when responses are sampled from $\pi_{\text{old}}$ rather than the current policy $\pi_\theta$.

### 2.2 REVISITING THE INSUFFICIENT EXPLORATION PROBLEM

We examine the exploration problem through entropy and performance changes on test and OOD benchmarks to characterize the issue and its implications. Figure 2 presents our analysis of GRPO's behavior during training on the MATH dataset. We observe two interconnected phenomena:

*(1) Rapid entropy collapse:* Despite incorporating substantial entropy regularization ($\lambda = 1 \times 10^{-3}$)*, the policy entropy decreases precipitously within the first few training steps, indicating rapid convergence to deterministic behaviors. This collapse stems from GRPO's inherently exploitative objective function (Equation 2), which prioritizes reward maximization over exploration.

*(2) Deteriorating generalization:* As entropy collapses, we observe a divergent trend: while test accuracy continues to improve, performance on OOD benchmarks such as AMC 23 declines. This suggests that reduced exploration causes the

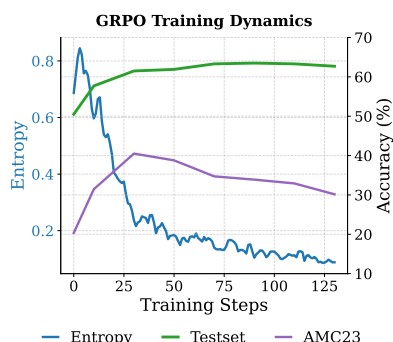

Figure 2: GRPO training dynamics: rapid entropy collapse accompanies rising Testset and decline on AMC23.

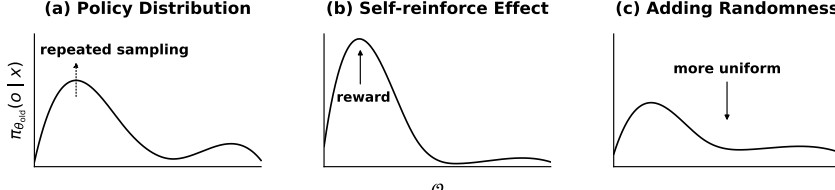

Figure 3: Illustration of exploration challenges in GRPO. (a) Policy distribution showing imbalanced modes with a dominant peak. (b) Self-reinforcement effect where the dominant mode becomes increasingly concentrated through positive feedback. (c) Effect of adding randomness (e.g., entropy regularization) which flattens the distribution but maintains the relative dominance of modes.

model to overfit to the training distribution rather than learn robust reasoning patterns that generalize to OOD benchmarks.

To explain entropy collapse, we hypothesize that when entropy begins to decline, the policy has formed partial, uncertain beliefs about the problem. In this regime, its response distribution contains multiple modes—multiple plausible reasoning traces can coexist for a given question. These modes are often imbalanced: one dominant mode accumulates a disproportionate share of probability mass, as illustrated in Figure 3(a). When responses are predominantly sampled from this dominant mode and receive positive feedback, the policy reinforces it further, amplifying its probability while suppressing alternative responses. The distribution evolves toward increasing imbalance, as shown in Figure 3(b). This self-reinforcing dynamic creates a feedback loop that inhibits exploration and ultimately leads to entropy collapse. This is particularly problematic: once a correct dominant mode emerges, it can prevent the discovery of alternative, potentially superior strategies, yielding local optima and limiting generalization to OOD benchmarks.

Current approaches to enhance exploration primarily increase randomness during optimization or sampling, such as strengthening the entropy term or raising the sampling temperature. These methods flatten the policy distribution toward a more uniform shape, as depicted in Figure 3(c). However, they do not disrupt the self-reinforcing loop: the dominant mode remains the most likely to be sampled even after flattening. This motivates our central question: *How can we enable the policy to explore plausible behaviors beyond the dominant mode during rollout?*

## 3 METHOD

### 3.1 EXPLORATION-ENHANCED POLICY OPTIMIZATION

To address the self-reinforcing dynamics that lead to entropy collapse, we propose Exploration-Enhanced Policy Optimization (EEPO), which prevents the rollout model from repeatedly sampling from dominant modes by *unlearning* previously sampled responses during rollout generation.

Figure 1 illustrates the key difference between GRPO and EEPO. In GRPO, the rollout model $\pi_{\text{rollout}}$ (corresponding to $\pi_{\text{old}}$ in Equation 2) samples all responses $O = \{o_1, o_2, \ldots, o_G\}$ from a fixed distribution, which are then used to compute rewards and advantages for policy optimization. EEPO introduces a sample-then-forget mechanism that divides the rollout into two stages separated by an unlearning step:

- *Stage 1 sampling:* Sample $G/2$ trajectories $\{o_1, o_2, \ldots, o_{G/2}\}$ from $\pi_{\text{rollout}}$.
- *Unlearning:* Update $\pi_{\text{rollout}}$ to forget the sampled trajectories.
- *Stage 2 sampling:* Sample the remaining trajectories $\{o_{G/2+1}, \ldots, o_G\}$ from the updated model.

After collecting all $G$ trajectories across both stages, we compute their rewards and apply the standard GRPO objective (Equation 2) to update the policy model. The denominator in Equation 3 uses the rollout model's probabilities, ensuring unbiased gradient estimates. Following standard GRPO practice, the rollout model is synchronized with the policy model at the beginning of each iteration, making the unlearning effect temporary and confined to the current rollout.

---

*This value is significantly larger than the $1 \times 10^{-4}$ suggested by SimpleRL (Zeng et al., 2025).

This approach decouples policy optimization from exploration: while the policy model $\pi_\theta$ focuses on reward maximization, the rollout model actively explores alternative trajectory spaces by suppressing previously visited regions. As illustrated in Figure 4, the unlearning step redistributes probability mass from dominant modes to other plausible regions, encouraging Stage 2 to sample from previously underexplored areas and effectively breaking the self-reinforcing loop that causes entropy collapse.

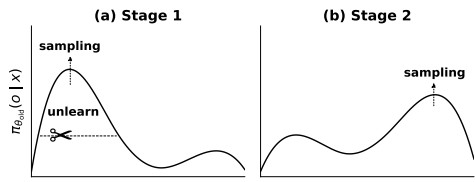

Figure 4: Unlearning suppresses the dominant mode and enables exploration of alternative modes that would otherwise be hard to reach.

### 3.2 ADAPTIVE UNLEARNING FOR ROLLOUT EXPLORATION

We now instantiate EEPO with an adaptive unlearning mechanism tailored to rollout-side exploration. The objective is to temporarily suppress dominant modes in $\pi_{\text{rollout}}$ when exploration begins to deteriorate. We identify three desiderata: (a) activate at the onset of entropy collapse to avoid disrupting healthy exploration, (b) penalize dominant regions more than others, and (c) remain lightweight and temporary. We realize these desiderata with three simple designs.

**Entropy-conditioned activation** To meet desideratum (a), we activate unlearning only during low-entropy phases; when entropy is high, no intervention is applied. We implement this via an entropy-based indicator:

$$\mathbb{I}_t = \mathbb{I}\left[\overline{\mathcal{H}}_t^{(m)} < \alpha\right], \tag{4}$$

where $\alpha > 0$ is a threshold and $\overline{\mathcal{H}}_t^{(m)}$ is the $m$-step moving average of token-level entropy at step $t$:

$$\overline{\mathcal{H}}_t^{(m)} = \frac{1}{m} \sum_{j=0}^{m-1} \mathcal{H}_{t-j}. \tag{5}$$

Here $\mathcal{H}_t$ denotes the token-level entropy at step $t$ (computed from $\pi_{\text{rollout}}(\cdot \mid q, o_{<t})$). A short horizon (e.g., $m = 3$) promptly detects low-entropy phases. This indicator multiplicatively gates the unlearning loss defined below.

**Complementary unlearning loss** To meet desideratum (b), unlearning strength should increase with prediction probability: strong in dominant regions with high probability mass and weak elsewhere. However, maximizing the standard *negative log-likelihood* (NLL) runs counter to our goal.

$$\mathcal{L}_{\text{NLL}} = -\log \pi_{\text{rollout}}(o_{k,t} \mid q, o_{k,<t}), \tag{6}$$

since it penalizes low-probability predictions more than high-probability ones (the loss goes to 0 as probability approaches 1). We therefore use a complementary loss that reverses this emphasis:

$$\mathcal{L}_{\text{Comp}} = \log\left(1 - \pi_{\text{rollout}}(o_{k,t} \mid q, o_{k,<t})\right), \tag{7}$$

which imposes stronger penalties on high-probability (dominant) predictions and weaker penalties on small-probability ones.

To ensure numerical stability as $\pi_{\text{rollout}}(o_{k,t}) \to 1$, we clip the probability before applying the loss:

$$p_{\text{clip}} = \min\left(\pi_{\text{rollout}}(o_{k,t} \mid q, o_{k,<t}), 1 - \epsilon\right), \tag{8}$$

where $\epsilon > 0$ is a small constant that prevents $1 - p_{\text{clip}}$ from approaching zero. The stabilized loss is:

$$\mathcal{L}_{\text{comp}} = -\log\left(1 - p_{\text{clip}}\right). \tag{9}$$

**Temporary single-step updates** To meet desideratum (c), we apply a single-step update to optimize the unlearning objective and confine its effect to the rollout model within each iteration. Let $o_k = (o_{k,1}, \ldots, o_{k,T_k})$ denote the $k$-th trajectory in the stage-1 rollout set $O_1 = \{o_1, o_2, \ldots, o_{G/2}\}$. The entropy-conditioned unlearning loss over $O_1$ is:

$$\mathcal{L}(O_1) = \frac{1}{|O_1|} \sum_{o_k \in O_1} \frac{1}{T_k} \sum_{t=1}^{T_k} \mathbb{I}_t \left[\log\left(1 - p_{\text{clip}}(o_{k,t})\right)\right]. \tag{10}$$

---

**Algorithm 1:** EEPO — Exploration-Enhanced Policy Optimization

---

**Initialize:** policy $\theta^0$; learning rates $\eta_{\text{GRPO}}, \eta$; group size $G$; iteration $K$; entropy threshold $\alpha$

**for** $k = 0$ to $K - 1$ **do**

    Sample $q \sim \mathcal{D}$; set $\theta' \leftarrow \theta^k$      // sample query and synchronize rollout from policy

    Sample $\{o_i\}_{i=1}^{G/2} \sim \pi_{\theta'}(\cdot \mid q)$      // Stage 1: sample $G/2$ trajectories

    **if** $\overline{\mathcal{H}}^{(m)}(\pi_{\theta'}) < \alpha$ **then**      // single-step adaptive unlearning

        $\theta' \leftarrow \theta' - \eta \, \nabla_{\theta'} \mathcal{L}(\{o_i\}_{i=1}^{G/2})$

    **end if**

    Sample $\{o_i\}_{i=G/2+1}^{G} \sim \pi_{\theta'}(\cdot \mid q)$      // Stage 2: sample remaining trajectories

    Form $O \leftarrow \{o_i\}_{i=1}^{G}$ and compute advantages $\{A(o)\}_{o \in O}$

    $\theta^{k+1} \leftarrow \theta^k + \eta_{\text{GRPO}} \, \nabla_\theta J_{\text{GRPO}}(\theta^k; O, r)$      // update policy with GRPO

**end for**

---

where $p_{\text{clip}}$ denotes the clipped probability and $\mathbb{I}_t$ is the entropy-based activation indicator. We then perform a single gradient ascend step without momentum to unlearn these trajectories:

$$\theta' \leftarrow \theta' + \eta \, \nabla_{\theta'} \mathcal{L}(\theta') , \tag{11}$$

where $\theta'$ parameterizes the rollout model, which is synchronized from the policy model (parameterized by $\theta$), $\theta' \leftarrow \theta$, as in GRPO's implementation (see Figure 1). Consequently, the unlearning effect is temporary—confined to the rollout model within the current iteration, without accumulation—and does not alter the policy parameters or optimization.

Algorithm 1 summarizes the EEPO procedure. It follows GRPO's structure but incorporates adaptive unlearning between the two rollout stages. After sampling the first $G/2$ trajectories (Stage 1), we check if policy entropy falls below threshold $\alpha$. If so, we perform a single gradient step to unlearn these trajectories using the complementary loss, temporarily modifying only the rollout model. We then sample the remaining $G/2$ trajectories (Stage 2) from the potentially modified rollout model. Finally, we update the policy with GRPO's objective on all $G$ trajectories. Note that in Eq. 3, the denominator is computed using the rollout model $\pi_{\theta'}$ that generated each trajectory.

## 4 EXPERIMENT

### 4.1 EXPERIMENTAL SETUP

**Datasets.** We train on the MATH dataset (Hendrycks et al., 2021a) using 8.5K hard problems (difficulty levels 3-5) following SimpleRL (Zeng et al., 2025). We evaluate on five mathematical reasoning benchmarks: Minerva Math (Lewkowycz et al., 2022), OlympiadBench (He et al., 2024), AMC 2023, AIME 2024. For the stronger Qwen3-8B-Base, we additionally include AIME 2025.

**Models.** We experiment with three LLMs: Qwen2.5-3B (Yang et al., 2024), Llama-3.2-3B-Instruct (Team, 2024), and Qwen2.5-7B-Instruct (Yang et al., 2024).

**Training Details.** We employ a binary reward (+1 for correct answer, 0 otherwise) without format constraints. All models are trained using VERL (Sheng et al., 2024) with GRPO for 2 epochs, using batch size 128, learning rate $5 \times 10^{-7}$, and 8 rollouts per question. For EEPO, we set entropy threshold $\alpha = 0.3$ and unlearning rate $\eta = 3 \times 10^{-3}$.

Further details of the experimental setup are provided in Appendix A.

### 4.2 BASELINES.

We compare EEPO to GRPO and other methods explicitly designed to enhance exploration.

**Base/Instruct Model.** The base model, or its instruction-tuned variant without additional reasoning-specific training, serving as performance lower bounds.

**GRPO.** GRPO applied to the base or instruction-tuned model using standard training settings.

**With Increased Entropy Term.** This variant encourages exploration by increasing the entropy weight in the objective function, prompting the actor to generate more diverse outputs.

**With Higher Sampling Temperature.** Applies a higher sampling temperature during actor's decoding process to promote exploration and reduce output determinism.

**With DAPO's Clip Higher.** Incorporates the "clip higher" technique from DAPO to encourage the selection of rare tokens during training.

**With More Rollouts.** Expands the exploration space by increasing the number of rollouts per training step, enabling broader trajectory sampling.

## 4.3 EXPERIMENTAL RESULTS

Table 1: Performance of EEPO compared to baselines on Qwen2.5-3B across four math benchmarks. Baseline results report the best performance across different hyperparameter settings (refer to Fig. 5). Average relative performance improvements (%) over GRPO are highlighted in blue.

| Method | Minerva Math | Olympiad Bench | AMC 23 | AIME 24 | Average |
|--------|--------------|----------------|--------|---------|---------|
| Qwen2.5-3B | 11.8 | 7.9 | 20.0 | 0.0 | 9.9 |
| GRPO | 22.4 | 27.9 | 30.3 | 3.3 | 21.0 |
| - Higher Temp. | 25.0 | 25.2 | 32.5 | 3.3 | 21.5 |
| - Increased Ent. | 25.0 | 29.6 | 37.5 | 3.3 | 23.9 |
| - DAPO Clip High. | 22.1 | 26.1 | 40.0 | 3.3 | 22.9 |
| - More rollouts. | 21.7 | 26.8 | 37.5 | 6.7 | 23.2 |
| EEPO | 23.5 **(+4.9)** | 29.3 **(+5.0)** | 45.0 **(+50.0)** | 6.7 **(+103.0)** | 26.1 **(+24.3)** |

**Overall results across three LLMs.** To validate the effectiveness of our method across different models and scales, we compare EEPO with baselines on three model families—Qwen2.5-3B, Llama3.2-3B-Instruct, and Qwen3-8B-Base. Tables 1–3 report the results. EEPO consistently outperforms GRPO and all exploration-enhanced GRPO variants across models and scales. Relative to standard GRPO, EEPO improves average accuracy by 24.3% on Qwen2.5-3B (21.0% → 26.1%), 33.0% on Llama3.2-3B-Instruct (17.6% → 23.4%), and 10.4% on Qwen3-8B-Base (34.7% → 38.3%). This pattern indicates that EEPO's sample-then-forget mechanism yields targeted exploration that scales from 3B to 8B parameters and transfers across base and instruction-tuned policies, providing a robust and model-agnostic improvement for mathematical reasoning under RLVR.

Table 2: Performance on Llama3.2-3B-Instruct.

| Method | Minerva Math | Olympiad Bench | AMC 23 | AIME 24 | Average |
|--------|--------------|----------------|--------|---------|---------|
| Llama3.2-3B-Instruct | 14.3 | 12.1 | 20.0 | 10.0 | 14.1 |
| GRPO | 19.5 | 17.5 | 20.0 | 13.3 | 17.6 |
| - Higher Temp. | 20.6 | 19.1 | 22.5 | 10.0 | 18.1 |
| - Increased Ent. | 20.2 | 18.1 | 30.0 | 10.0 | 19.6 |
| - DAPO Clip High. | 19.1 | 17.3 | 25.0 | 16.7 | 19.5 |
| - More rollouts. | 19.1 | 17.2 | 22.5 | 16.7 | 18.9 |
| EEPO | 20.6 **(+5.6)** | 18.1 **(+3.4)** | 35.0 **(+75.0)** | 20.0 **(+50.4)** | 23.4 **(+33.0)** |

**Comparison with baselines.** We compare EEPO to four exploration strategies, each evaluated at its best hyperparameter setting (Figure 5). Despite careful tuning, all baselines fail to match EEPO's performance. While these strategies can outperform GRPO, gains are modest and require brittle tuning. Temperature-based exploration exhibits a clear exploration–exploitation trade-off: performance peaks around 1.2 but degrades sharply at higher values (1.5). We also observe substantially longer training time at the best temperatures (1.2) due to the much longer reasoning paths caused by inefficient exploration (Figure 8). Clip-higher and entropy regularization likewise swing between under- and over-exploration and lag behind EEPO across all models. Increasing the number of rollouts provides benefits but plateaus quickly while computational cost also grows substantially (Figure 8). In contrast, EEPO achieves larger gains by enabling targeted exploration within the rollout process.

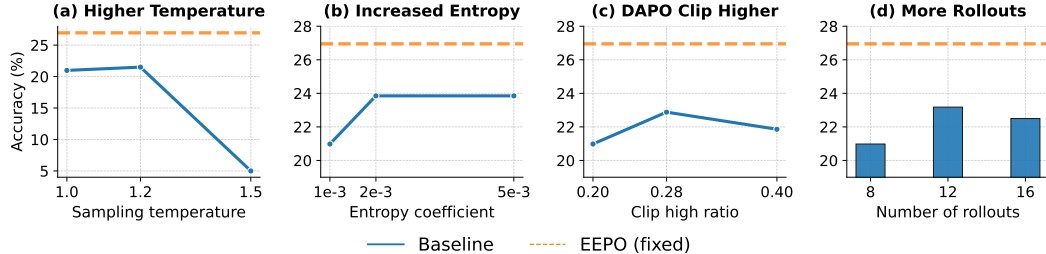

Figure 5: Impact of hyperparameter choices on baselines performance using Qwen2.5-3B. Each subplot shows the average accuracy across four math benchmarks as a function of (a) temperature, (b) entropy coefficient, (c) clip higher ratio, and (d) number of rollouts. The orange dashed line represents the EEPO with fixed hyperparameters.

**Generalization to benchmarks.** To assess generalization, we evaluate EEPO against baselines on five diverse math reasoning benchmarks, as shown in Tables 1–3. Our method achieves consistent improvements over GRPO across all benchmarks. Performance continues to improve on harder and distribution-shifted splits where baselines plateau. On a competition-level benchmark with Qwen2.5-3B, EEPO reaches 45.0% compared to 30.3% for GRPO. These gains stem from EEPO's sustained exploration and superior entropy maintenance (Figures 6 and 7), which prevent the entropy collapse that leads to overfitting on the training distribution and degraded generalization (Figure 2).

Table 3: Performance on Qwen3-8B-Base.

| Method | Minerva Math | Olympiad Bench | AMC 23 | AIME 24 | AIME 25 | Average |
|---|---|---|---|---|---|---|
| Qwen3-8B-Base | 33.1 | 36.0 | 52.5 | 10 | 13.3 | 29.0 |
| GRPO | 41.2 | 45.5 | 50.0 | 20.0 | 16.6 | 34.7 |
| - Higher Temp. | 40.1 | 44.3 | 55.0 | 16.7 | 20.0 | 35.22 |
| - Increased Ent. | 40.4 | 42.8 | 60.0 | 16.7 | 20.0 | 35.9 |
| - DAPO Clip High. | 40.1 | 41.6 | 55.0 | 16.7 | 10.0 | 32.7 |
| - More rollouts. | 40.8 | 44.0 | 57.5 | 16.7 | 16.7 | 35.1 |
| EEPO | 41.5 | 44.3 | 62.5 | 20.0 | 23.3 | 38.3 **(+10.4)** |

## 5 ANALYSIS

**Effectiveness of EEPO: Exploration Enhancement and Quality Preservation.** To understand the effectiveness of EEPO, we compare its training dynamics with GRPO, as shown in Figure 6.

The entropy dynamics in Figure 6(a) reveal how sample-then-forget changes exploration behavior. While GRPO exhibits continuous entropy collapse indicating that responses samples increasingly concentrate on high-probability modes, EEPO maintains consistently higher entropy throughout training. Notably, EEPO's Stage 2 achieves higher entropy than Stage 1, suggesting that temporary response suppression successfully forces the model to explore low-density regions that the original actor rarely visits. This entropy gap demonstrates that our mechanism effectively prevents mode collapse by strategically sampling from diverse regions of the probability distribution.

Despite this enhanced exploration, generation quality remains preserved. Figure 6(b-c) shows that both mean rewards and response lengths of EEPO remain stable and comparable to GRPO. These results validate our hypothesis: temporarily suppressing sampled responses can enhance exploration by steering the actor away from high-probability regions toward other plausible alternatives, while preserving the generation capabilities necessary for effective training.

**Generalization and Pass@k.** Figure 7 shows that EEPO delivers better generalization dynamics and higher final performance on AMC23 (Figure 7a), with improved Pass@k scaling as the sampling budget increases (Figure 7b). By mitigating entropy collapse (see Figure 6a) and maintaining higher

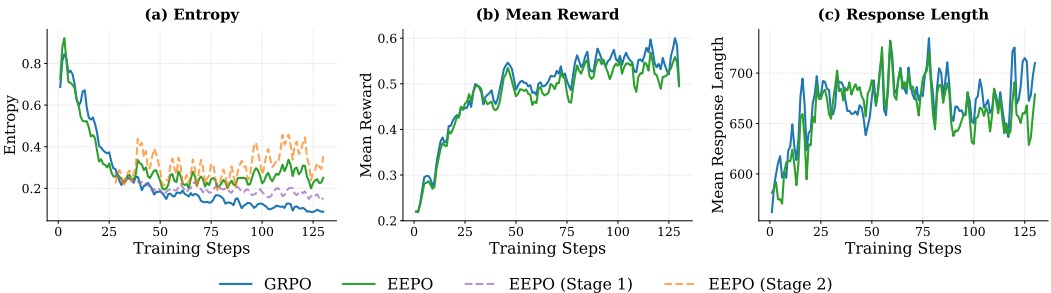

Figure 6: Training dynamics comparison between EEPO and GRPO. (a) Entropy evolution shows EEPO maintains higher exploration ability throughout training, with Stage 2 exhibiting increased entropy compared to Stage 1, demonstrating effective exploration enhancement through 'sample-then-forget' mechanism. In contrast, GRPO exhibits monotonic entropy decay. (b) Mean reward trajectories remain comparable between methods and across EEPO stages. (c) Response length distributions show similar patterns, indicating preserved generation quality.

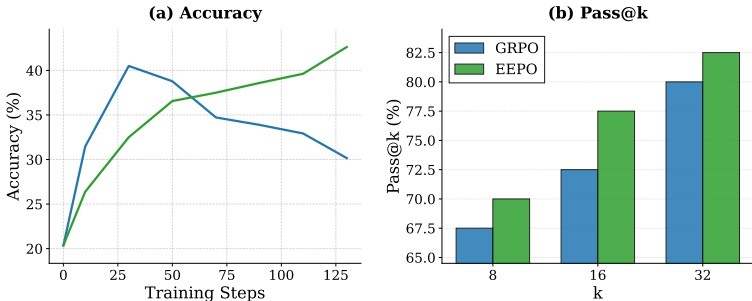

Figure 7: Performance comparison of GRPO and EEPO on AMC23 benchmark using Qwen2.5-3B. (a) Training accuracy dynamics; (b) Pass@k scaling with sampling budgets. EEPO achieves higher final performance and better scaling with increased computation.

policy entropy, EEPO continues to sample non-dominant yet plausible modes, sustaining exploration throughout training. This stabilized exploration prevents overfitting to the training distribution and discovers reasoning patterns that generalize to the OOD benchmark AMC23, while also yielding improvements in Pass@k under larger sampling budgets.

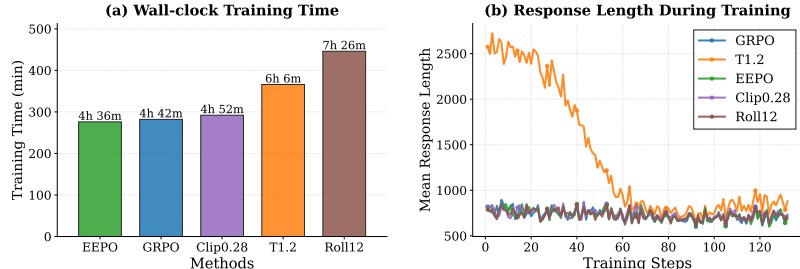

Figure 8: Training efficiency comparison on Qwen3-8B-Base. (a) Wall-clock training time for EEPO and baseline methods. (b) Mean response length during training for each method. EEPO achieves the fastest training time while maintaining stable response lengths.

**Training Efficiency.** We evaluate the computational efficiency of EEPO and baseline methods on Qwen3-8B-Base using B200 GPUs. As shown in Figure 8(a), EEPO achieves comparable training time to standard GRPO, demonstrating that our exploration mechanism introduces negligible computational overhead within the entire framework. Among baseline configurations, higher sampling temperatures significantly slow training by approximately 30%, as these methods generate substantially longer responses throughout training (Figure 8(b)). Additional rollouts incur the highest computational cost due to increased trajectory sampling, while adjusting the clipping ratio has

minimal impact on efficiency. These results demonstrate that EEPO achieves superior performance through effective exploration while preserving the training efficiency of the original GRPO algorithm.

## 6 RELATED WORK

**Reinforcement learning with verifiable rewards.** Reinforcement learning has been widely used to improve the capabilities of language models (Bai et al., 2022; Rafailov et al., 2023; Wang et al., 2025; Zhang et al., 2026b) and to align model outputs with human values under diverse scenarios (Zhang et al., 2026a; Chen et al., 2025c;b), particularly through reinforcement learning from human feedback (RLHF) (Ouyang et al., 2022). More recently, reinforcement learning with verifiable rewards (RLVR) (Shao et al., 2024; DeepSeek-AI et al., 2025; Team et al., 2025) replaces subjective preference feedback with automatically verifiable, rule-based rewards, enabling scalable optimization in domains with reliable checkers, such as mathematics (Hendrycks et al., 2021b; Chen et al., 2025a) and programming (Codeforces, 2025; Jin et al., 2026). For example, DeepSeek-R1 (DeepSeek-AI et al., 2025) reports that RLVR can elicit a range of reasoning behaviors—including verification and self-correction—often accompanied by longer intermediate reasoning traces (Gandhi et al., 2025). However, RLVR can be unstable and may plateau early: insufficient exploration can cause premature convergence to narrow solution modes, limiting further improvement.

**Exploration in RL.** Exploration in RL is often promoted through policy stochasticity under the assumption that randomness broadens coverage of actions and states; however, indiscriminate randomness is insufficient, as policies tend to collapse toward near-deterministic behavior—"entropy collapse" (Cui et al., 2025; Yu et al., 2025)—driven by exploitative objectives. Recent efforts largely fall into two categories: objective-level modifications and indiscriminate exploration. The latter increases randomness uniformly, for example via $\epsilon$-greedy policies (Sutton & Barto, 2018), softmax temperature adjustments (Chen et al., 2025d; Hou et al., 2025), or entropy regularization (Hou et al., 2025); while these methods raise stochasticity, they do not shift probability mass away from dominant behaviors and often become unstable or ineffective when applied aggressively. On the objective side, increasing the clipping threshold (e.g., DAPO (Yu et al., 2025)) or concurrent work's relaxing rewards with Pass@k (Chen et al., 2025e) admits more low-probability trajectories but leaves rollout dynamics unchanged, allowing the policy to repeatedly sample high-probability regions and sustain the self-reinforcing loop (§ 2.2) that drives entropy collapse. In contrast, we propose a active rollout-time intervention that temporarily forgets recently sampled trajectories, explicitly discouraging revisits and steering the model to explore alternative modes in sequence; this targeted mechanism disrupts self-reinforcement and remains complementary to objective-level adjustments.

**Machine Unlearning for LLMs** Machine unlearning for LLMs studies removing the influence of specific data (e.g., sensitive or copyrighted content) without retraining models from scratch (Liu et al., 2024). Typical motivations include privacy compliance and mitigating bias or harmful behaviors. Common approaches involve weight editing (Mitchell et al., 2022) or gradient-based optimization (Jang et al., 2023) to forget targeted data, and inference-time strategies such as prompt manipulation. However, prior work primarily focuses on knowledge erasure, whereas EEPO repurposes and tailors unlearning for RL exploration: during rollout generation, we temporarily unlearn previously sampled trajectories to prevent the rollout model from repeatedly sampling from dominant modes.

## 7 CONCLUSION

We introduced EEPO, an exploration-enhanced policy optimization framework that augments the rollout process with a sample-then-forget mechanism. By temporarily suppressing recently sampled trajectories during rollouts, EEPO encourages exploration of alternative modes in the output distribution that would otherwise remain underexplored. Our method transforms indiscriminate stochasticity into strategic exploration, breaking the self-reinforcing loop that causes insufficient exploration and entropy collapse. Extensive experiments across three models and five mathematical reasoning benchmarks demonstrate that EEPO consistently outperforms existing methods while maintaining comparable training efficiency. These results establish EEPO as a practical and effective approach for addressing the exploration-exploitation trade-off in RLVR.

## ACKNOWLEDGMENTS

We thank the anonymous reviewers for their constructive feedback and suggestions. This work was partially supported by Hong Kong RGC GRF No. 14206324. QZW and BH were supported by RGC General Research Fund No. 12200725.

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

## LLM USAGE DISCLOSURE

In our work, we mainly use GPT-5 for writing enhancements, primarily to improve grammar and text clarity.

## ETHICS STATEMENT

All authors have read and adhered to the ICLR Code of Ethics. Our study relies solely on publicly available datasets and models, as detailed in Appendix A. No private or personally identifiable information was used. The work aims to advance the scientific understanding of PO methods while upholding principles of transparency, fairness, and responsible research.

## REPRODUCIBILITY STATEMENT

The codebase will be made publicly available upon acceptance. The base models and benchmarks used in this work are publicly accessible. All experiments were conducted using NVIDIA A100 80GB GPUs and B200 184G GPUs.

## A   DETAILED EXPERIMENTAL SETUP

**Datasets.**   We use the MATH dataset (Hendrycks et al., 2021a) for RL training. Following the setup of SimpleRL (Zeng et al., 2025), we train on the hard data, which contains 8.5K problems with difficulty levels ranging from 3 to 5. For evaluation, we adopt five challenging mathematical reasoning benchmarks: Minerva Math (Lewkowycz et al., 2022), OlympiadBench (He et al., 2024), and three recent competition-level datasets—AMC 2023, AIME 2024, and AIME 2025. For smaller models (Qwen2.5-3B and LLaMA-3.2-3B-Instruct), evaluation is conducted on the first four benchmarks. For the stronger Qwen3-8B-Base model, we additionally include AIME 2025.

**Models.**   To demonstrate the generality of our approach, we experiment with three LLMs from different model families and scales.

- Qwen2.5-3B (Yang et al., 2024): a base model from the Qwen2.5 series, with stronger pretraining and support for long-context inputs.
- Llama-3.2-3B-Instruct (Team, 2024): an instruction-following model based on Meta's Llama architecture, included to evaluate cross-family generalization.
- Qwen3-8B-Base (Yang et al., 2025): a larger base model from the Qwen3 family, used to assess performance at a larger scale.

**Reward Function.**   We employ a binary reward based on answer correctness: +1 for a correct final answer and 0 otherwise. We exclude format-based rewards, which can constrain exploration and degrade performance (Zeng et al., 2025), particularly when training base models.

**Implementation Details.**   All models are trained using the VERL framework (Sheng et al., 2024), employing the GRPO algorithm. We use a batch size of 128, a mini-batch size of 64, a learning rate of $5 \times 10^{-7}$, and 8 rollouts, training for 2 epochs. The KL loss and entropy loss coefficient are set to $1 \times 10^{-4}$ and $1 \times 10^{-5}$, respectively. The maximum response length varies by model: up to 4K tokens for Qwen2.5-3B, and up to 6K tokens for both LLaMA-3.2-3B-Instruct and Qwen3-8B-Base. During evaluation, we use greedy decoding to compute pass@1 accuracy. All experiments are conducted on compute clusters equipped with NVIDIA A100 GPUs (80GB) and B200 GPUs.

## B   BASELINES DESCRIPTION

We compare EEPO against GRPO and several variants that are explicitly designed to enhance exploration.

**Base/Instruct Model.** The base model, or its instruction-tuned variant without any additional reasoning-specific training, serves as a performance lower bound.

**GRPO.** Standard GRPO applied to the base or instruction-tuned model using default training settings.

**Increased Entropy Regularization.** This variant enhances exploration by increasing the entropy weight in the training objective, encouraging the policy to generate more diverse outputs. It represents a common approach where stronger entropy regularization is used to promote exploration.

**Higher Sampling Temperature.** This variant applies a higher sampling temperature during the actor's decoding process to promote exploration and reduce output determinism. Temperature-based softmax exploration (also known as Boltzmann exploration) is a widely used method to implement the $\varepsilon$-greedy algorithm in stochastic policies. As the temperature $t \to 0$, the policy becomes nearly greedy; as $t \to \infty$, the action distribution approaches uniform, effectively increasing exploration.

**Clip Higher.** This variant incorporates the "clip higher" heuristic from DAPO, which encourages the selection of rare or low-probability tokens during training. It is one of the most widely used exploration-enhancing baselines in modern RLVR pipelines.

**Increased Number of Rollouts.** This baseline increases the number of rollouts per training step to expand the trajectory space and encourage broader exploration. It is designed to evaluate whether EEPO with 8 rollouts can match or outperform GRPO with a larger number of rollouts (default: 16).

## C    EXPERIMENTS ON LARGE-SCALE MODELS

To assess how EEPO scales with model size, we extend our experiments from 3B and 8B models to a larger 14B model, Qwen3-14B-Base. The results are summarized in Table 4.

Table 4: Results on Qwen3-14B-Base across five reasoning benchmarks.

| Method | Benchmark | | | | | Avg. |
|---|---|---|---|---|---|---|
| | Minerva Math | OlympiadBench | AMC23 | AIME24 | AIME25 | |
| GRPO | 36.8 | 48.6 | 67.5 | 23.3 | 26.7 | 40.6 |
| EEPO | 39.3 | 50.1 | 67.5 | 36.7 | 30.0 | 44.7 |

As shown in Table 4, EEPO continues to provide consistent improvements over GRPO on Qwen3-14B-Base, particularly on the more challenging benchmarks (e.g., AIME24 and AIME25). This suggests that EEPO scales well with model size and remains effective in the 3B–14B range.

## D    ADDITIONAL COMPARISON WITH GRPO VARIANTS

To make the gain of EEPO more directly and comparable, we also provide the following fair comparisons, where EEPO is s implemented on GRPO and its variants.

| Method | Avg. acc. |
|---|---|
| GRPO | 21.0 |
| EEPO | 26.1 |
| GRPO + Increased Entropy | 23.9 |
| EEPO + Increased Entropy | 27.9 |
| GRPO + Clip High | 22.9 |
| EEPO + Clip High | 26.6 |

Table 5: Average accuracy of GRPO variants and their EEPO-enhanced counterparts. EEPO provides gains of 3.7–5.1 absolute accuracy points over already strong exploration-enhanced baselines.

Table 5 reports the average accuracy of GRPO and its variants, together with their EEPO-enhanced counterparts. EEPO consistently yields an absolute improvement of about 3.7–5.1 points over the corresponding exploration-enhanced GRPO methods.

