# OpenReview forum: "EEPO: Exploration-Enhanced Policy Optimization via Sample-Then-Forget"
_ICLR.cc/2026/Conference — ICLR 2026 Poster_

### Official Review · Reviewer_X4NZ · 2025-10-20

**Soundness:** 3
**Presentation:** 3
**Contribution:** 3
**Rating:** 6
**Confidence:** 4

**Summary:**

This paper addresses the challenge of insufficient exploration and resulting entropy collapse in Reinforcement Learning with Verifiable Rewards (RLVR) used for training large language model (LLM) reasoning capabilities. Current methods often over-exploit dominant trajectories, and while adding randomness increases stochasticity, it fails to effectively shift the policy away from these modes. To counter this, the authors propose Exploration-Enhanced Policy Optimization (EEPO), a framework that modifies the rollout phase with a novel "sample-then-forget" mechanism. EEPO performs rollouts in two stages: after sampling the first half of trajectories, it applies a temporary, lightweight unlearning step to the rollout model to suppress the just-sampled responses, using a complementary loss function gated by an entropy threshold. This forces the second stage of sampling to explore different, less probable regions of the output space, actively steering away from dominant modes. The standard GRPO objective is then used to update the main policy model using the combined trajectories from both stages, decoupling the exploration enhancement from the policy optimization itself. Experiments across five reasoning benchmarks and three different LLMs show that EEPO consistently outperforms GRPO and other exploration-focused baselines, achieving significant average accuracy gains.

**Strengths:**

1. The paper investigates the critical problem of insufficient exploration and entropy collapse in Reinforcement Learning with Verifiable Rewards (RLVR), a central challenge that limits the performance gains and generalization of LLMs trained for reasoning tasks.
2. The proposed Exploration-Enhanced Policy Optimization (EEPO) method is conceptually intuitive, employing a straightforward "sample-then-forget" mechanism within a two-stage rollout process to actively suppress dominant modes and encourage exploration.
3. The effectiveness and robustness of EEPO are demonstrated through comprehensive experiments conducted across five different mathematical reasoning benchmarks and three distinct LLM architectures and sizes.

**Weaknesses:**

1. The comparison primarily focuses on standard GRPO and variations based on randomness or rollout count, neglecting direct benchmarks against more advanced, contemporary RLVR algorithms like the full DAPO framework.
2. Experimental validation is confined to LLMs with fewer than 10 billion parameters (specifically 3B and 8B models), leaving the scalability and effectiveness of EEPO on larger foundation models untested.
3. While EEPO aims to improve performance via enhanced exploration, the provided training curves show its mean reward is comparable to or even slightly lower than standard GRPO (Fig. 6), and the paper lacks a detailed explanation for why the proposed method results in slightly faster wall-clock training times despite incorporating an additional unlearning step.

**Questions:**

Please see paper weaknesses.

---

> ### Author Response · Authors · 2025-11-21
>
> Thank you for the positive and constructive assessment. We appreciate your recognition of EEPO as a novel and intuitive method targeting a central challenge in RLVR. Your note on the significant performance gains across comprehensive experiments is very encouraging. We address the concerns point by point below.
>
>
> ## Response to W1: Comparison with the DAPO framework
>
> We appreciate the suggestion and have compared EEPO with DAPO on three models: Qwen2.5-3B, Llama3.2-3B-Instruct, and Qwen3-8B-Base.
>
> Table 1. Results on Qwen2.5-3B.
> | Qwen2.5-3B | minerva math | olympiad bench | amc23 | aime24 | AVG |
> | :--- | :--- | :--- | :--- | :--- | :--- |
> | DAPO | 22.8 | 27.5 | 35.0 | 6.7 | 23.0 |
> | EEPO | 23.5 | 29.3 | 45.0 | 6.7 | 26.1 |
>
> Table 2. Results on Llama3.2-3B-Instruct.
> | Llama3.2-3B-Instruct | minerva math | olympiad bench | amc23 | aime24 | AVG |
> | :--- | :--- | :--- | :--- | :--- | :--- |
> | DAPO | 18.8 | 18.1 | 25.0 | 13.3 | 18.8 |
> | EEPO | 20.6 | 18.1 | 35.0 | 20.0 | 23.4 |
>
> Table 3. Results on Qwen3-8B-Base.
> | Qwen3-8B | minerva math | olympiad bench | amc23 | aime24 | aime25 | AVG |
> | :--- | :--- | :--- | :--- | :--- | :--- | :--- |
> | DAPO | 40.1 | 43.1 | 62.5 | 13.3 | 16.7 | 35.1 |
> | EEPO | 41.5 | 44.3 | 62.5 | 20.0 | 23.3 | 38.3 |
>
> As shown in Tables 1–3, EEPO outperforms DAPO across all models and benchmarks. These results further support the effectiveness of our approach. The corresponding numbers have been incorporated into the main results Tables 1-3 of section 4.3 in the revised manuscript.
>
>
> ## Response to W2: Experiments on models ≥10B
>
> Thank you for the suggestion to evaluate larger models. In response, we extended our experiments to Qwen3-14B-Base, going beyond the 3B and 8B settings.
>
> Table 4. Results on Qwen3-14B-Base.
> | Qwen3-14B | minerva math | olympiad bench | amc23 | aime24 | aime25 | AVG |
> | :--- | :--- | :--- | :--- | :--- | :--- | :--- |
> | GRPO | 36.8 | 48.6 | 67.5 | 23.3 | 26.7 | 40.6 |
> | EEPO | 39.3 | 50.1 | 67.5 | 36.7 | 30.0 | 44.7 |
>
> As shown in Table 4, EEPO continues to provide improvements over GRPO on 14B Qwen, particularly on the more challenging benchmarks. This suggests that EEPO scales well with model size and remains effective in the 3B–14B range. The results are now included in the appendix C.
>
>
> ## Response to W3: Explanation of reward curves and training time
>
> We appreciate the question regarding the interpretation of Fig. 6 and the training time. We provide explanations as follows:
>
> **Exploration vs. mean training reward.**
> EEPO is designed to improve *exploration* of RLVR, which is measured by the entropy metric in Fig. 6a. In contrast, the training rewards in Fig. 6b captures *exploitation* of RL on trajectories that yield high rewards. For example, if the policy repeatedly samples the same correct trajectory, the mean training reward approaches 1 while the entropy becomes very low. Such entropy collapse can hurt generalization, as illustrated in Fig. 1 of the paper.
>
> Taken together, Figs. 6a and 6b show that EEPO enhances exploration (higher entropy) while maintaining a comparable level of exploitation (mean training reward), achieving a better exploration–exploitation trade-off and improved generalization.
>
> **Wall-clock training time.**
> The overall wall-clock training time of EEPO is comparable to that of GRPO, largely due to a slight reduction in mean response length under EEPO (Fig. 6c), which lowers trajectory-generation cost. This reduction offsets the additional unlearning computation, leading to similar training times. We have added this explanation to the revised analysis section.
>
> ***
>
> Thank you again for the detailed and insightful review. Your comments helped us strengthen the work. We have revised the paper accordingly, with changes highlighted in *green* for clarity. We hope the additional experiments and analyses address your concerns, and we welcome any further feedback.

---

> > ### Comment · Reviewer_X4NZ · 2025-11-24
> >
> > Thanks for your responses. My major concerns are addressed, and I would like to raise my score to accept. I recommend the authors include the extended experiments on DAPO and 14B-LLM into their final version of paper (if accepted). Also, the analysis on wall-clock time can be briefly integrated into somewhere in the main text.

---

> > > ### Author Response · Authors · 2025-11-24
> > > **Thank you for your support**
> > >
> > > Thank you very much for raising your score and for your constructive suggestions. We are delighted that our revisions addressed your concerns. We will incorporate all these improvements into the final version.
> > >
> > > We sincerely appreciate your thoughtful engagement throughout the review process.

---

### Official Review · Reviewer_WrTx · 2025-10-26

**Soundness:** 3
**Presentation:** 2
**Contribution:** 3
**Rating:** 6
**Confidence:** 2

**Summary:**

This work introduces EEPO, a framework that addresses the exploration-exploitation dilemma in RLVR. EEPO enhances the rollout process through a sample-then-forget mechanism, which temporarily suppresses recently sampled trajectories during generation. This deliberate exclusion discourages the policy from repeatedly collapsing into dominant, high-probability modes and instead promotes the discovery of alternative, underexplored reasoning pathways. By converting passive stochasticity into purposeful, diversity-driven exploration, EEPO steers the policy toward more comprehensive coverage of the output distribution.

Extensive experiments across three model families and five mathematical reasoning benchmarks show that EEPO outperforms existing methods in final performance while preserving comparable training efficiency.

**Strengths:**

(1) Well-written (2) Detailed experiment (3) The problems related to training efficiency that have been solved are distinctive and seem valuable to the industrial sector

**Weaknesses:**

N/A

**Questions:**

I do not know this specialized research field very well. I will adjust my score and optimize my review document based on the evaluations of other expert reviewers and my performance during the rebuttal period.

---

> ### Author Response · Authors · 2025-11-21
>
> We sincerely thank you for the positive and encouraging assessment. We appreciate your recognition of the clarity of our writing, the thoroughness of our experiments, and the practical value of our work to the industrial sector. We are grateful for the time and effort you dedicated to reviewing our submission, and we welcome any further feedback you may have.

---

### Official Review · Reviewer_zNJJ · 2025-10-27

**Soundness:** 3
**Presentation:** 3
**Contribution:** 2
**Rating:** 4
**Confidence:** 2

**Summary:**

The authors introduce a novel method to maintain exploration while learning in LLMs with RLVR. Current approaches overemphasize exploitation, causing premature entropy policy collapse. The current method avoids this by adaptive unlearning of the dominant modes and focusing on secondary modes. Improving performance compared to baselines is shown.

**Strengths:**

The problem solution is simple and elegant. This is complemented by simple and useful intuitions that help us to understand the proposed algorithm.

**Weaknesses:**

Nice intuitions are presented, like in Fig. 2, but no theory is provided to back up those intuitions. This means that improvement can strongly depend on parameters and on a case-by-case basis.

Ablations studies are missing, so it is unclear what aspect of the introduced algorithm is critical: is it enough to have an entropy-conditioned activation of unlearning just based on Eq. 8, assuming a constant $p_{clip}$?

Is it possible that in some problems no secondary mode exists? Then how would the method fare in discovering the most promising actions in the vast space of tokens and reasoning trajectories?

While performance increases compared to other methods, the performance is very close to GRPO with updated parameters, so it seems that the gains are relatively small.

As far as I can see, no comparison with other methods of RL exploration is introduced (epsilon-greedy with optimal epsilon, entropy regularization, and such). Does the method outperform the obvious modifications of the algorithms with those additional exploration tricks?

**Questions:**

See weaknesses.

---

> ### Author Response · Authors · 2025-11-21
> **Rebuttal by Authors [1/3]**
>
> Thank you for the detailed and constructive assessment. We appreciate your highlighting of EEPO as a novel and elegant method, complemented by useful and nice intuition, and your recognition that it improves performance over baselines. We address the comment point by point below.
>
>
> ## Response to “Nice intuitions are presented, but no theory is provided”
>
> Thank you for pointing this out. In response, we provide a theoretical analysis to support the intuitions in Fig. 2 and EEPO in Appendix. F, `Self-Reinforcement Effect and How EEPO Counters It`. Below we summarize the main results.
>
>
>
> ### **RL updates are intrinsically self-reinforcing**
> We first analyze a standard RLVR objective with a softmax head. For a query, let the model assign probabilities $p_i^t$ to candidates and let $r_i \in \{0,1\}$ be a reward indicator. The expected reward is $J = \sum_i r_i p_i^t$. A gradient-ascent step on the logits $z^t$ (with $p^t = \mathrm{Softmax}(z^t)$ and step size $\eta' > 0$ absorbing the feature norm) yields:
>
> $$
> z_k^{t+1} =
> \\begin{cases}
> z_k^t + \\eta' p_k^t (1-\\bar r^t), & r_k = 1, \\\\
> z_k^t - \\eta' p_k^t \\bar r^t, & r_k = 0,
> \\end{cases}
> \\quad
> \\bar r^t = \\sum_i r_i p_i^t.
> $$
>
>
> **Lemma 1.** For any two positive modes $i,j$ (i.e., $r_i = r_j = 1$), we have $\frac{p_i^{t+1}}{p_j^{t+1}} = \frac{p_i^t}{p_j^t} \cdot \exp\big(\eta'(1 - \bar r^t)(p_i^t - p_j^t)\big)$. If $p_i^t > p_j^t$ and $\bar r^t < 1$, then $\frac{p_i^{t+1}}{p_j^{t+1}} > \frac{p_i^t}{p_j^t}$. So, **the more probable positive mode becomes strictly more dominant after one RL update**.
>
> This shows RL is inherently mode-seeking and explains the self-reinforcement effect in Fig. 2(a–b).
>
>
>
> ### **Complementary unlearning is a mode-favoring mass transport**
>
> To characterize *where the “removed” probability mass goes during unlearning*, we analyze the gradient flow induced by EEPO’s complementary loss.
>
> For a fixed query and a selected mode $y$, we consider $\\frac{d z_k}{d\\tau} = -\\frac{\\partial \\mathcal{L}_{\\text{comp}}}{\\partial z_k}$, and $p(\\tau) = \\mathrm{Softmax}(z(\\tau))$.
>
> Using the explicit gradient of $\\mathcal{L}_{\\text{comp}}$, we obtain the logit flow:  $\\frac{d z_y}{d\\tau} = -p_y$,  $\\frac{d z_k}{d\\tau} = \\frac{p_y p_k}{1 - p_y},\\ (k \\neq y)$.
>
> Combined with $\\frac{d p_k}{d\\tau} = p_k\\left(\\frac{d z_k}{d\\tau} - \\sum_u p_u \\frac{d z_u}{d\\tau}\\right)$, this yields closed-form ODEs. Let $S_1 = \\sum_{u \\neq y} p_u = 1 - p_y$,  $S_2 = \\sum_{u \\neq y} p_u^2$. Then $\\frac{d p_y}{d\\tau} = -p_y^2\\left(1 - p_y + \\frac{S_2}{S_1}\\right) < 0$, and $\\frac{d p_k}{d\\tau} = p_k p_y\\left(\\frac{p_k}{1 - p_y} + p_y - \\frac{S_2}{S_1}\\right) \\quad (k \\neq y)$.
>
>
> **Lemma 2.** For any unlearning step on a sampled mode $y$, we have $\frac{d p_y}{d\tau} < 0$ and, for all $k \neq y$, $\frac{d}{d\tau}\left(\frac{p_k}{p_y}\right) > 0$. Thus, the selected mode $y$ always loses probability mass, and every ratio $p_k / p_y$ strictly increases.
>
> To understand how mass is redistributed among the non-selected modes, define the fractional growth rate $\gamma_k \triangleq \frac{1}{p_k} \frac{d p_k}{d\tau}$ for $k \neq y$. We prove:
>
> **Lemma 3.** For any $i,j \neq y$, $\gamma_i - \gamma_j = \frac{p_y}{1 - p_y}(p_i - p_j)$. Since $p_y > 0$ and $1 - p_y > 0$, the sign of $\gamma_i - \gamma_j$ equals the sign of $p_i - p_j$. Thus, **among non-selected modes, higher-probability ones always grow faster.**
>
> **Corollary 4.** Let $i^\star \in \arg\max_{k \neq y} p_k$. Whenever $p_y > 0$, $\frac{d p_{i^\star}}{d\tau} > 0$. So, **the top alternative always gains mass. The transport isn’t uniform or random — it’s mode-favoring**.
>
>
> Taken together, Lemma 1–3 and Corollary 4 show that **RL updates amplify mode dominance**, while **EEPO's complementary unlearning performs a mode-favoring mass transport**: it pushes down the sampled mode and shifts probability toward already plausible alternatives (especially the top one), rather than adding random noise. This provides a theoretical explanation for the dynamics in Fig. 2 and supports the design of EEPO.

---

> > ### Comment · Reviewer_zNJJ · 2025-11-21
> >
> > I thank the authors for their constructive and detailed responses.
> > I still think that the paper does not provide solid theory. For instance, a simple MDP problem could be used in discrete action and state space to show provable convergence in that case. The current work is mostly showing empirical results, but there is not solid theory that back them up. I did not check the new math provided in detail yet, but I think that it does not provide any convergence proof -- or proof that other modes would be found in a simple MDP problem -- when combining anti self-reinforcement with self-recinforcing. Correct me if I am wrong.

---

> ### Author Response · Authors · 2025-11-21
> **Rebuttal by Authors [2/3]**
>
> ## Response to 'Ablations studies are missing'
>
> Thank you for the comment. The most important components of EEPO are (i) the unlearning step and (ii) the entropy-conditioned activation. We have added ablation studies for both, and we also explain the role of the clipping operation.
>
>
>
> **Clipping for numerical stability.**
> The clipping parameter is a small fixed constant introduced solely for numerical stability. Without clipping, probabilities $p$ very close to 1 can cause numerical issues when computing the unlearning loss term (e.g., $\log(1 - p)$). We therefore clip $p$ away from 1 to avoid numerical instability. Here, $p$ denotes the token-level probability distribution from the model, and $p_{\mathrm{clip}}$ is its clipped version used for computation stability.
>
>
> **Entropy threshold $\alpha$.**
> The entropy threshold $\alpha$ controls when the policy entropy is sufficiently low that additional exploration should be encouraged. We have added an ablation over $\alpha \in \{0.0, 0.1, 0.2, 0.3, 0.4\}$. Here, $\alpha = 0.0$ corresponds to GRPO (no intervention).
>
> Table 1. Ablation on $\alpha$.
> | $\alpha$  | 0.0 (GRPO) | 0.1  | 0.2  | 0.3  | 0.4  |
> | :-------- | :--------- | :--- | :--- | :--- | :--- |
> | avg. acc. | 21.0       | 25.2 | 24.8 | 26.1 | 25.4 |
>
> As shown in Table 1, EEPO consistently improves over GRPO across a reasonably wide range of $\alpha$, with the best performance at $\alpha = 0.3$.
>
>
> **Unlearning learning rate $\eta$.**
> The unlearning learning rate $\eta$ controls the step size of the unlearning update. We have added an ablation over  $\eta \in \{0, 1\times10^{-4}, 1\times10^{-3}, 3\times10^{-3}, 1\times10^{-2}\}$. The case $\eta = 0$ again reduces to GRPO (no unlearning).
>
>
> Table 2. Ablation on $\eta$.
>
> | $\eta$ | 0 (GRPO) | $1 \mathrm{e}-4$ | $1 \mathrm{e}-3$ | $3 \mathrm{e}-3$ | $1 \mathrm{e}-2$ |
> | :--- | :--- | :--- | :--- | :--- | :--- |
> | avg acc| 21.0 | 23.3 | 24.4 | 26.1 | 22.5 |
>
> As Table 2 shows, performance improves steadily as $\eta$ increases up to $3\times10^{-3}$, while an overly large rate ($10^{-2}$) makes the unlearning step unstable and degrades performance, which is consistent with intuition. The corresponding results and explanations are now included in Appendix D.
>
>
>
> ## Response to “Is it possible that in some problems no secondary mode exists?”
>
> Thank you for this insightful question. While such cases are possible, we believe they are very rare in practice, as in high-dimensional reasoning spaces it is unlikely for the output distribution to have a single sharp peak and be flat elsewhere. Below we discuss how EEPO behaves in this scenario across the two rollout stages.
>
> First, in EEPO the *unlearning step is applied only to the rollout model*, while the GRPO actor remains unchanged and is always updated with the standard GRPO objective using all collected trajectories. Consequently, even if there is effectively a single peak mode, rollouts drawn before unlearning (Stage-1) still sample this mode as in standard GRPO.
>
> Second, if the output distribution does have a single peak and is flat elsewhere, then suppressing this dominant mode in the rollout model essentially redistributes probability mass to the remaining regions, making the distribution closer to uniform. In this special case, Stage-2 rollouts behave similarly to using a higher sampling temperature: exploration is mildly increased and serves only as a complement to the Stage-1 rollouts from the original rollout distribution.
>
> Overall, when no meaningful secondary mode exists, EEPO effectively reduces to GRPO with slightly more exploratory rollouts, without degrading the dominant high-reward mode.

---

> ### Author Response · Authors · 2025-11-21
> **Rebuttal by Authors [3/3]**
>
> ## Response to 'the performance is very close to GRPO with updated parameters'
>
> Thank you for this question, and we apologize for any confusion caused by our earlier presentation. We clarify that in our setting, EEPO and GRPO are directly comparable: EEPO is built on top of the GRPO setup (we keep the shared hyperparameters consistent).
>
> When we refer to “GRPO with updated parameters”, it actually corresponds to *GRPO plus exploration methods*, e.g., with increased entropy regulation, with more rollouts — *GRPO's 16 rollouts vs. EEPO’s 8 rollouts*, not the same base that EEPO is implemented on.
>
> To make the gain of EEPO more direct, we now provide the following fair comparisons, where EEPO is implemented on GRPO and its variants. In this setting, EEPO achieves an absolute accuracy gain of about 3.7–5.1 points:
>
>    | Method                    | avg. acc. |
>    | :------------------------ | :-------- |
>    | GRPO                      | 21.0      |
>    | EEPO                      | 26.1      |
>
>    | Method                    | avg. acc. |
>    | :------------------------ | :-------- |
>    | GRPO + Increased Entropy  | 23.9      |
>    | EEPO + Increased Entropy  | 27.9      |
>
>    | Method                    | avg. acc. |
>    | :------------------------ | :-------- |
>    | GRPO + Clip High          | 22.9      |
>    | EEPO + Clip High          | 26.6      |
>
> We consider these improvements substantial, especially given that the exploration-enhanced baselines are already strong. These results and the accompanying explanation have been added to the appendix E.
>
>
>
> ## Response to 'no comparison with other methods of RL exploration is introduced (epsilon-greedy with optimal epsilon, entropy regularization, and such).'
>
>
> Thank you for raising this point. We would like to clarify that these exploration methods are indeed included in our experiments (Tables 1–3 in the manuscript), albeit under slightly different names:
>
> - *Entropy regularization*: this corresponds to the *“Increased Entropy”* baseline in our experiments, where we add an entropy bonus term to the objective.
>
> - *Epsilon-greedy*:  this corresponds to the *“Higher Temp”* baseline. Temperature-based softmax exploration (also Boltzmann exploration) is a standard way to trade off greedy exploitation and uniform exploration. As the temperature $t \rightarrow 0$, the policy becomes nearly greedy; as $t \rightarrow \infty$ , the distribution approaches uniform.
>
> We also perform hyperparameter sweeps to select optimal hyperparameters for these baselines; the corresponding curves are shown in Fig. 5. We have added these explanations in the section 4.2 to make it clearer.
>
>
> ***
>
>
> Thank you once again for your insightful comments, which have helped us strengthen the work. We have updated the manuscript accordingly and highlighted the changes in *blue and orange* for easy reference. We hope that our responses and revisions could effectively address your concerns, and we welcome any further feedback you may have.

---

> ### Author Response · Authors · 2025-11-24
>
> Thank you very much for your follow-up comment and for clarifying what kind of theory you had in mind. Our previous theory mainly focused on analyzing the _dynamics_ of the self-reinforcement effect of RL and how unlearning breaks this effect, rather than on providing a convergence-style guarantee.
>
> In the revised manuscript, we have added a convergence analysis for the policy update under EEPO in Appendix G (“Convergence of EEPO’s Policy Update”). We summarize the main result here.
>
> ## Convergence analysis of EEPO
>
> EEPO modifies only the _rollout generation_ process, while the policy $\pi_\theta$ is always updated by an importance-weighted GRPO objective that aggregates all collected trajectories. This can be regarded as adding slightly off-policy samples and correcting the distribution mismatch with importance sampling. Based on standard stochastic gradient descent analysis for non-convex optimization, we show that EEPO's policy update converges to a stationary point at rate $O(1/\sqrt{T})$, where $T$ is the number of outer iterations.
>
>
>
> **Theorem (Convergence of EEPO policy update)**
> *Suppose the objective $J \in \mathcal{J}_n$, where $\mathcal{J}_n$ is the class of finite-sum $L$-smooth functions, has $\sigma$-bounded gradients, and the importance weights are clipped by w_max. With step size $\eta_t = c/\sqrt{T}$ where*
>
> $$c = \sqrt{2(J(\boldsymbol{\theta}^\star) - J(\boldsymbol{\theta}^0))/(L w_{\max}^2 \sigma^2)}, $$
>
> *we have*
>
>
> $$ \min_{0 \le t \le T-1} \mathbb{E}\left[\|\nabla J(\boldsymbol{\theta}^t)\|^2\right] \le w_{\max} \sigma \sqrt{ \frac{2L \left( J(\boldsymbol{\theta}^\star) - J(\boldsymbol{\theta}^0) \right)}{T} } = O\left(\frac{1}{\sqrt{T}}\right) $$
>
>
>
> This result depends only on the fact that EEPO's two-stage rollouts are properly reweighted by the corresponding importance ratios $\pi_{\boldsymbol{\theta}}(\tau) / \pi_{\text{roll}}^{(t)}(\tau)$.  The unlearning step only changes the rollout distribution $\pi_{\text{roll}}^{(t)}$ (and hence the distribution of trajectories), but it does not change the form of the policy update.
>
>
> ***
>
> In summary, while our primary focus is empirical, we provide two lines of theoretical support for EEPO:
>
> **(i) Dynamics analysis (Appendix F).** A gradient-flow analysis that explains the self-reinforcement / mode-seeking effect of RL updates illustrated in Fig. 2, and how unlearning breaks this effect via mode-favoring mass transport.
> **(ii) Convergence analysis (Appendix G).** A convergence result showing that EEPO’s importance-weighted policy update reaches a stationary point at the rate of $O(1/\sqrt{T})$.
>
>
> Together with our previous responses (which addressed ablations, baselines, and the initial dynamics analysis), we hope this help address your concerns. We would be happy to clarify further if needed.

---

> ### Author Response · Authors · 2025-11-27
> **Brief follow-up on our response**
>
> Dear Reviewer zNJJ,
>
> Thank you again for your review of our paper and for your follow-up comment. In response, we have added a convergence analysis of EEPO’s policy update in Appendix G, in addition to the earlier dynamics analysis in Appendix F.
>
> We would appreciate it if you could let us know whether this addition and our previous responses (detailing ablations, baselines, numerical results, and the initial dynamics analysis) help address your concerns, and if any further information or clarification is needed.
>
> Best regards,
>
> Authors

---

### Official Review · Reviewer_SHnD · 2025-10-31

**Soundness:** 2
**Presentation:** 3
**Contribution:** 2
**Rating:** 4
**Confidence:** 3

**Summary:**

The authors proposes Exploration-Enhanced Policy Optimization (EEPO), a novel framework for reinforcement learning with verifiable rewards (RLVR) that addresses policy entropy collapse.

**Strengths:**

1. The "sample-then-forget" idea is a clever and novel approach to the exploration-exploitation problem. Instead of just adding indiscriminate noise (like increasing temperature), it actively and strategically steers the policy away from dominant modes it is starting to overfit on.

2. The method demonstrates consistent and significant performance gains over strong GRPO baselines across five challenging mathematical reasoning benchmarks and three different language models. The average relative improvements are substantial (e.g., +33.0% on Llama3.2-3B-Instruct).

**Weaknesses:**

1. The method introduces at least two key new hyperparameters: the entropy threshold $\alpha$ for activating unlearning and the unlearning rate $\eta$. The paper uses fixed values ($\alpha=0.3$, $\eta=3\times10^{-3}$) without providing an ablation study or discussion on how these values were chosen or how sensitive the model's performance is to them. This could be a point of fragility.

2. The unlearning step modifies the rollout policy ($\pi_{\theta^{\prime}}$) in the middle of generating a single batch of trajectories. It is unclear how the importance sampling (IS) ratio for the standard GRPO objective (Eq. 2) remains valid when trajectories in the same batch ($O$) are drawn from two different policies (pre-unlearning and post-unlearning). This potential violation of the IS assumption needs a more rigorous justification.

**Questions:**

Please address the concerns in weakness section.

---

> ### Author Response · Authors · 2025-11-21
>
> Thank you for the constructive feedback, and for highlighting EEPO as a clever and novel approach that achieves consistent and significant gains over GRPO. We address the comments below.
>
>
> ## Response to W1: Ablation of hyperparameters
> **Entropy threshold $\alpha$.**
> The entropy threshold $\alpha$ controls when the policy entropy is sufficiently low that additional exploration should be encouraged. In practice, we selected $\alpha$ by inspecting the training curves  (Fig. 1), where we observed the following:
>
> - The generalization performance begins to degrade when the entropy enters roughly the $[0.2, 0.4]$ range, with a tipping point around $0.3$.
> - Before this range, entropy decays rapidly; afterward, the decay becomes much flatter, indicating that the policy has already become highly concentrated.
>
> We therefore set $\alpha = 0.3$.
>
> Table 1. Ablation on $\alpha$.
> | $\alpha$  | 0.0 (GRPO) | 0.1  | 0.2  | 0.3  | 0.4  |
> | :-------- | :--------- | :--- | :--- | :--- | :--- |
> | avg. acc. | 21.0       | 25.2 | 24.8 | 26.1 | 25.4 |
>
> We additionally performed an ablation study over $\alpha \in \{0.0, 0.1, 0.2, 0.3, 0.4\}$. Here, $\alpha = 0.0$ corresponds to GRPO (no intervention). As shown in Table 1, EEPO demonstrates improvements over GRPO across a reasonably wide range of $\alpha$.
>
> **Unlearning learning rate $\eta$.**
> The unlearning learning rate $\eta$ controls the step size of the unlearning update. In practice, we choose $\eta$ to be as large as possible while keeping the unlearning process stable.
>
> Table 2. Ablation on $\eta$.
>
> | $\eta$ | 0 (GRPO) | $1 \mathrm{e}-4$ | $1 \mathrm{e}-3$ | $3 \mathrm{e}-3$ | $1 \mathrm{e}-2$ |
> | :--- | :--- | :--- | :--- | :--- | :--- |
> | avg acc| 21.0 | 23.3 | 24.4 | 26.1 | 22.5 |
>
> We also added an ablation over $\eta \in \{0, 1\times10^{-4}, 1\times10^{-3}, 3\times10^{-3}, 1\times10^{-2}\}$. The case $\eta = 0$ again reduces to GRPO (no unlearning). Performance improves steadily as $\eta$ increases, while an overly large rate ($10^{-2}$) makes the unlearning step unstable and degrades performance, which is consistent with intuition.
>
> The corresponding results and explanations are now included in  Appendix D.
>
>
> ## Response to W2: Calculation of the IS ratio
>
> We appreciate your thoughtful question and apologize for any confusion caused by our earlier presentation. In our implementation, the denominator $\pi_{\theta_{\text{old}}}$ in the importance sampling (IS) ratio is always taken to be the rollout model $\pi_{\theta'}$ that actually generated each sample. This ensures that the IS estimator remains unbiased under the standard assumptions.
>
> Formally, for each trajectory $\tau$, we compute:
>
> $$
> \text{IS}(\tau) = \frac{\pi_{\theta}(\tau)}{\pi_{\theta'}(\tau)},
> $$
>
> where $\pi_{\theta}$ is the current policy and $\pi_{\theta'}$ is the rollout model used for that specific sample.
>
> We have clarified this in the method section of the revised manuscript. Thank you for highlighting this point.
>
> ***
>
> Thank you once again for your insightful comments, which have strengthened our work. We have updated the manuscript to reflect these changes (highlighted in orange for easy identification). We hope that our responses and revisions could effectively address your concerns, and we welcome any further feedback you may have.

---

> ### Author Response · Authors · 2025-11-24
> **Brief follow-up on our response**
>
> Dear Reviewer SHnD,
>
> Thank you again for your insightful questions. Following your suggestions, we have
> (i) added ablations for the entropy threshold and unlearning rate (Appendix D), and
> (ii) clarified the importance sampling ratio calculation in the method section.
>
> If you have a chance to take another look, we would be very grateful, and we would be happy to clarify anything further if needed.
>
> Best regards,
>
> Authors

---

> ### Author Response · Authors · 2025-11-27
> **Gentle reminder: Request for your feedback**
>
> Dear Reviewer SHnD,
>
> Thank you for your time and efforts in reviewing our work and for your insightful questions. Following your comments, we have provided detailed responses and revised the manuscript accordingly.
>
> Please let us know whether our responses address your concerns, and if any further clarification is needed.
>
> Best regards,
>
> Authors

---

### Author Response · Authors · 2025-11-21
**General Response**

Dear Reviewers,

We sincerely appreciate your time and expertise in reviewing our paper. We are pleased that EEPO is perceived as a novel, intuitive, clever, and elegant approach (SHnD, zNJJ, X4NZ), supported by nice and useful intuitions (zNJJ). We also appreciate the recognition that it targets a central challenge in RLVR, which limits the performance and generalization of reasoning models (X4NZ), and that it is practically valuable for the industrial sector (WrTx). We are further encouraged by your comments that EEPO achieves significant performance gains across five benchmarks and three different LLMs (SHnD, WrTx, X4NZ). Additionally, we are grateful that the writing and presentation of the manuscript were well received (SHnD, zNJJ, WrTx, X4NZ).

We believe that all of the reviewers' concerns can be addressed. Below, we provide brief responses to the main concerns and suggestions raised in the reviews:

**Ablation of hyperparameters (SHnD, zNJJ).**   We have added discussion and ablations of the key hyperparameters — the entropy threshold $\alpha$ and the unlearning learning rate $\eta$ — in Appendix D, illustrating that EEPO is robust across a reasonably wide range of settings.

**Experiments on models ≥10B and DAPO baseline (X4NZ, zNJJ).**  We have added experiments on Qwen3-14B-Base and included the DAPO baseline. These results are reported in Appendix C and in Tables 1–3 of Section 4.3. We have also clarified the baseline configurations and comparisons in Section 4.2 and Appendix E.

**Theoretical analysis (zNJJ).** To support the intuitions in Fig. 2 and the design of EEPO, we have provided a detailed theoretical analysis in Appendix F (“Self-Reinforcement Effect and How EEPO Counters It”), showing both (i) the self-reinforcing, mode-seeking nature of standard RL updates and (ii) how EEPO’s unlearning step induces a mode-favoring mass transport toward alternative high-probability modes.

All of these revisions have been incorporated into the revised manuscript and are highlighted for your convenience.

We hope that our responses and revisions adequately address your concerns, and we would be very grateful if you could take them into account when making your final evaluation of our work.

Sincerely,
Authors

---

### Meta-Review · Area_Chair_HxD4 · 2026-01-06

**Summary:**

Here is a summary of the reviewers' concerns:
- Limited empirical justification: there is no comparison with other methods of RL exploration (epsilon-greedy with optimal epsilon, entropy regularization, etc), and more advanced, contemporary RLVR algorithms like the full DAPO framework..
- Unclear scalability: Experimental validation is confined to LLMs with fewer than 10 billion parameters (specifically 3B and 8B models), leaving the scalability and effectiveness of EEPO on larger foundation models untested.
- Lack ablation study and discussion on new hyper-parameters
- Potential violation of the importance sampling assumption
- How to handle uni-mode problems
- Lack deep analysis of experimental results:  training curves show its mean reward is comparable to or even slightly lower than standard GRPO (Fig. 6), and the paper lacks a detailed explanation for why the proposed method results in slightly faster wall-clock training times despite incorporating an additional unlearning step.
- Lack theoretical justification: no theory is provided to back up the intuitions shown in Fig. 2; did not provide convergence analysis.
- Limited performance gain comparing to GRPO with updated parameters.

**Reviewer Concerns:**

Here are the reviewer concerns addressed by the rebuttal:
- Ablation study
- Calculation of the IS ratio
- Explained how to handle uni-mode problems
- performance gain comparing to GRPO
- Limited empirical justification: added the experiments required by reviewers.
- Scalability to larger model size
- Explanation of reward curves and training time

The concerns that are partially addressed:
- Theoretical justification

**Reviewer Scores:**

Reviewer SHnD raised 2 concerns, most of the concerns were addressed by the rebuttal. Reviewer SHnD might increase the score to 6.

Reviewer zNJJ raised 5 concerns, most of his concerns were addressed fully or partially by the rebuttal. Reviewer zNJJ might keep the score due to partially addressed theoretical justification.

Reviewer WrTx did not raise any concerns, and might keep the score.

Reviewer X4NZ raised 3 concerns, the authors addressed them fully. Reviewer X4NZ might raise the score to 8.

[Note] the paper exceeds the 9 page limit.

---

### Decision · Program_Chairs · 2026-01-26

Accept (Poster)